# FSH-DETR: An Efficient End-to-End Fire Smoke and Human Detection Based on a Deformable DEtection TRansformer (DETR)

**DOI:** 10.3390/s24134077

**Published:** 2024-06-23

**Authors:** Tianyu Liang, Guigen Zeng

**Affiliations:** 1School of Computer Science, Nanjing University of Posts and Telecommunications, Nanjing 210023, China; b21111324@njupt.edu.cn; 2School of Communications and Information Engineering, Nanjing University of Posts and Telecommunications, Nanjing 210023, China; 3Telecommunication and Networks National Engineering Research Center, Nanjing University of Posts and Telecommunications, Nanjing 210003, China

**Keywords:** fire smoke and human detection, Deformable-DETR, Mixed Encoder, PIoUV2, ConvNeXt

## Abstract

Fire is a significant security threat that can lead to casualties, property damage, and environmental damage. Despite the availability of object-detection algorithms, challenges persist in detecting fires, smoke, and humans. These challenges include poor performance in detecting small fires and smoke, as well as a high computational cost, which limits deployments. In this paper, we propose an end-to-end object detector for fire, smoke, and human detection based on Deformable DETR (DEtection TRansformer) called FSH-DETR. To effectively process multi-scale fire and smoke features, we propose a novel Mixed Encoder, which integrates SSFI (Separate Single-scale Feature Interaction Module) and CCFM (CNN-based Cross-scale Feature Fusion Module) for multi-scale fire, smoke, and human feature fusion. Furthermore, we enhance the convergence speed of FSH-DETR by incorporating a bounding box loss function called PIoUv2 (Powerful Intersection of Union), which improves the precision of fire, smoke, and human detection. Extensive experiments on the public dataset demonstrate that the proposed method surpasses state-of-the-art methods in terms of the mAP (mean Average Precision), with mAP and ${mAP}_{50}$ reaching 66.7% and 84.2%, respectively.

## 1. Introduction

Accidental fires in our daily lives can cause harm to personal and property safety. According to the National Fire Protection Association, in 2022, the US fire department responded to an estimated 1.5 million fires, which resulted in 3790 civilian deaths, 13,250 civilian injuries, and an estimated $18 billion in property damage [1]. At the same time, the damage caused by fires to the natural environment cannot be ignored. In 2023, the total area burned by wildfires in Canada exceeded 156,000 square kilometers, exceeding the benchmark established in 1995. The record-breaking fire released airborne pollutants and greenhouse gases, contributing significantly to climate alteration [2]. In the event of a fire, it is of the utmost importance to act promptly. The timely detection of a fire and its victims can effectively reduce the harm. Traditional fire alarm systems, such as photoionization smoke detectors, infrared thermal imagers, flame gas sensors, and smoke gas sensors, have inherent limitations, including delayed response times and restricted sensor densities. Especially in open spaces, airflow and other conditions may impede accurate detection [3].

Early visualization-based systems for detecting fire, smoke, and humans involve techniques, such as color detection, moving-object detection, and motion and flicker analysis using Fourier and wavelet transforms, among others [4]. Dalal et al. introduced a texture-based method that counted the occurrences of the gradient orientation in localized ports of an image, computed on a dense grid of uniformly spaced cells and used overlapping local contrast normalization for human detection [5]. P. V. Koerich Borgesf et al. achieved fire detection by evaluating the inter-frame variations of features, such as color, area size, and texture, in potential fire zones and combining Bayesian classifiers [6]. Yusuf Hakan Habiboğlu et al. proposed a flame-detection system that employs a spatiotemporal covariance matrix of video data, which effectively captures the flickering and irregular characteristics of flames by dividing the video into spatiotemporal blocks and calculating the covariance features extracted from these blocks [7]. Although numerous physical and mathematical methods have been used to extract features, such as the color, texture, and flicker frequency contour of fire, smoke, and humans, these early methods have been constrained by their limited feature representation capability due to their manually designed feature extractors. Furthermore, they have demonstrated poor adaptability to complex scene changes, dynamic backgrounds, and lighting modifications, resulting in elevated missed detection rates and weak generalization ability [8].

The rapid development and increasing maturity of neural networks have led to the emergence of Convolutional Neural Networks (CNNs). As a dominant force in the field of computer vision, CNNs have demonstrated a remarkable capacity for extracting rich and discriminative features from extensive data [8,9,10], a capability that has attracted the attention of a vast number of researchers. Object-detection algorithms based on CNNs are increasingly applied for fire, smoke, and human detection [3,8,9,10,11]. According to different processing procedures and structures, they can be broadly classified into two categories: one-stage algorithms and two-stage algorithms. One-stage methods directly estimate the object location and category from input images, thereby eliminating the need for detecting potential target regions beforehand. These algorithms operate by dividing the image into grids, generating diverse bounding boxes based on anchor points in each grid, and employing non-maximum suppression (NMS) [12] to eliminate redundant and overlapped bounding boxes. The representative of one-stage algorithms is the You Only Look Once (YOLO) series [13,14,15,16,17,18]. Two-stage algorithms complete object-detection tasks through two main stages: candidate box generation and object detection. Initially, a component called candidate box generators, such as Selective Search [19] or Region Proposal Network [20], is employed to generate potential target-containing candidate boxes that are produced in the input image. Subsequently, these candidate boxes undergo filtering and feature extraction using NMS, followed by classification and regression within classification and regression heads. Algorithms, such as Fast R-CNN [21], Faster R-CNN [20], Cascade R-CNN [22], and Sparse R-CNN [23], exemplify this category. Although two-stage algorithms exhibit superior precision relative to one-stage methods, they often have higher hardware requirements due to their high computational complexity and are challenging to meet real-time requirements [24].

A novel object-detection method, DEtection TRansformer (DETR), has recently emerged for object detection, achieving excellent results comparable to the mature Faster R-CNN on the COCO dataset [25]. Inspired by the transformer architecture, which was initially adopted in fields like natural language processing and speech recognition, DETR showcases substantial advancements. DETR firstly enables end-to-end object detection, meaning it directly predicts the bounding box coordinates and class labels without relying on anchor boxes or region proposal techniques. This simplifies the object-detection pipeline and eliminates the need for complex components, like NMS, anchor generation, and anchor matching. The end-to-end nature of DETR makes it more efficient and easier to implement compared to traditional algorithms. Zhu, X. et al. have made improvements to DETR and proposed a new model called Deformable DETR. Compared with DETR, Deformable DETR has better detection performance, lower computational complexity, and faster convergence. It is worth noting that Deformable DETR performs exceptionally well in detecting small target objects [26]. In the early stages of a fire, smoke and fire tend to be concentrated in a small area [27]. The advantage of Deformable DETR in detecting small objects is helpful in the timely detection of small flames and smoke, which can prevent the fire from spreading. Additionally, Deformable DETR introduces the concept of Deformable Convolution [28], which selects only a few points near the reference point as k in self-attention calculation. This approach not only speeds up the convergence of the model but also improves its computational efficiency, allowing it to detect irregular flames and smoke more effectively. In the past, fire detection often overlooked the detection of humans. Adding people as detection objects in fire and smoke detection tasks is of great significance for firefighters to promptly rescue victims.

Nevertheless, the utilization of Deformable DETR for object detection continues to be confronted with considerable obstacles. Although Deformable DETR shows excellent prediction precision based on the COCO dataset, it is not satisfied with real-time tasks in terms of the computational cost and inference speed. To address these issues, we have made several improvements. First, the original ResNet [29] is replaced by an advanced ConvNeXt, which enhances the network’s capacity to extract complex features related to fire, smoke, and humans. Secondly, the high computational cost of the encoder part of Deformable DETR renders it unsuitable for deployment on resource-constrained detection devices. To simplify its structure and enhance the detection precision, we have implemented modifications to the encoder part. Third, the GIoU (Generalized Intersection over Union) [30] in the Deformable DETR limits the convergence speed and detection precision, and therefore, Powerful IoU (PIoU) v2 is introduced as a new loss function. Our contributions can be summarized as follows:(1)We propose FSH-DETR for the precise and rapid detection of fire, smoke, and humans. In response to complex and dynamic fire environments, we introduce ConvNeXt to enhance the algorithm’s ability to extract features of varying scales.(2)To improve detection precision and significantly reduce computational costs, we propose the Mixed Encoder, which integrates SSFI (Separate Single-scale Feature Interaction Module) and CCFM (CNN-based Cross-scale Feature Fusion Module) [31].(3)To solve the issue of slow convergence and improve the model’s stability in complex fire scenarios, we introduce PIoU v2 as the loss function.(4)Extensive experiments on the public dataset have demonstrated that our model achieves superior detection precision with less computational cost compared to the baseline.

This paper is structured as follows. In Section 2, we review related works and discuss their strengths and limitations. Section 3 details the overall architecture and improvement methods of our proposed model. Section 4 introduces the experimental setup, including the dataset, evaluation methods, and experimental environment. In Section 5, to demonstrate the detection performance and characteristics of our model, visual examples, qualitative analysis, and comparisons with other methods are provided. Section 6 summarizes the entire study and provides prospects for future work.

## 2. Related Works

One-stage algorithms: Given the fast inference speed and low hardware requirements of one-stage algorithms, most fire and human detection tasks prefer this type of algorithm. Nguyen et al. achieved real-time human detection by adjusting the input size, output size, and residual blocks of YOLOV2 and adding Spatial Pyramid Pooling blocks [32]. Valikhujaev et al. proposed a new model for fire and smoke detection based on dilated convolution to overcome limitations, such as unusual camera angles and seasonal variations [33]. Mukhiddinov et al. implemented an improved YOLOv5 drone image detection system for wildfire smoke. They improved the backbone of the network using a spatial pyramid pooling fast plus layer and applied a bidirectional feature pyramid network for easier access and faster multi-scale feature fusion [34]. Saydirasulovich et al. used Wise IoU v3 for bounding box regression, Ghost Shuffle Convolution for parameter reduction, and the BiFormer attention mechanism to capture the characteristics of forest fire smoke. The model they proposed solved the problems of poor detection precision and the difficulty in distinguishing small-scale smoke sources in wildfire smoke detection [35]. Ergasheva et al. enhanced the dataset using histogram equalization technology and successfully developed an effective early detection model for ship fires based on YOLOV8 [36]. Although one-stage algorithms are simple and fast and can achieve real-time object detection, their detection precision is still not as good as some two-stage algorithms [37]. Meanwhile, the YOLO series is not ideal for detecting small target objects [38], making it naturally disadvantageous in detecting early fire characteristics.

Two-stage algorithms: In contrast to one-stage detectors that focus on speed, two-stage detectors focus on precision. To address the crowding occlusion problem, Kevin Zhang et al. proposed Double Anchor R-CNN, which utilized Double Anchor RPN and a proposal crossover strategy to generate and effectively aggregate proposals. Finally, a Joint NMS is introduced to improve the stability of post-processing [39]. P Barmpoutis et al. introduced a fire-detection approach integrating deep learning networks and linear dynamic systems. Initially, the Faster R-CNN network detected potential fire regions within the image. Then, the regions were projected onto the Grassmannian space. Finally, a vector of indigenous aggregated descriptors was used to group Grassmannian points. [40]. Chaoxia et al. advanced the anchor formulation strategy of Faster R-CNN using the color-guided anchoring strategy, while simultaneously constructing a Global Information Network (GIN) to obtain global image information, enhancing the efficiency and precision of flame detection [41]. Pan J et al. used a knowledge distillation process to make Faster R-CNN lightweight and proposed a weakly supervised fine-segmentation method for detection and classification. A fuzzy system was introduced to construct a fire and smoke rating framework [37]. Nevertheless, mainstream two-staged methods show poor precision in small-object detection [38]. More critically, anchor-based methods, like Faster R-CNN, face challenges in locating objects with diverse shapes [42], which is a drawback for detecting amorphous fire and smoke.

DETR-based algorithms: One-stage and two-stage algorithms are mostly anchor-based methods. According to recent research, the detection performance of anchor-based algorithms depends to some extent on the initial value of the set number of anchors [43]. Both too many and too few anchors lead to poor results, and excessive anchors also increase computational complexity. Unfortunately, these algorithms use NMS during the detection process, rather than all edge devices supporting NMS (such as edge computing devices that only support integer operations) [44]. In order to solve the above problems and abandon manual intervention and the application of prior knowledge, researchers have begun to turn their attention to transformer-based DETR. Matthieu Lin et al. proposed a new decoder DQRF and a faster bipartite matching algorithm, successfully applying DETR to pedestrian detection [45]. Li, Y. et al. applied lightweight DETR in fire and smoke detection, reducing the number of encoder layers and incorporating a multi-scale deformable attention mechanism. They also used ResNeXt50 as the backbone and added the normalization-based attention module (NAM) to improve the model’s feature-extraction ability [46]. Mardani, K. et al. simplified DETR by removing unnecessary components, such as binary matching and bounding box heads, and added masked or linear layers composed of Multi-head attention layers to complete different tasks, achieving optimal precision performance based on specified datasets [47]. Huang, J. et al. used Deformable DETR as the baseline and combined a Multi-scale Context Controlled Local Feature Module (MCCL) and Dense Pyramid Pooling Module (DPPM) to improve the ability of small smoke detection [10].

Recent improvements to DETR have mainly focused on improving the decoder section. For instance, Conditional DETR decouples the cross-attention function of the DETR decoder and proposes conditional spatial embedding, which accelerates the model’s convergence speed [48]. Dynamic Anchor Box DETR (DAB-DETR) uses dynamically updated box coordinates as queries in the decoder, achieving the goal of improving the model precision and convergence speed [49]. New research indicates that low-scale features account for 75% of all tokens in the encoder, but they make a small contribution to the overall detection precision [50]. Therefore, we focus on improving the rarely studied encoder block in this article. Compared with the baseline (Deformable DETR), we reduce the number of encoder layers from six to two, decreasing the computational cost. Simultaneously, Separate Self-Attention and CCFM are employed to substitute for the Multi-scale Deformable Attention [26] in the encoder block. Finally, we replaced the backbone with ConvNeXt, a more advanced architecture with stronger feature extraction capabilities than the traditional ResNet.

## 3. Methodology

### 3.1. Overall Architecture of FSH-DETR

FSH-DETR (Fire Smoke and Human detection based on Deformable DETR) is a transformer-based object detection algorithm for detecting fire, smoke, and humans. As shown in Figure 1, FSH-DETR shares a similar network structure with DETR, comprising three main components: a backbone for feature extraction, an encoder-decoder transformer for locating objects, and a feed-forward network (FFN) for predicting results. Upon entering the model, the image undergoes initial feature extraction via the backbone, followed by advanced feature extraction via our proposed Mixed Encoder. The Mixed Encoder module consists of a Separate Single-scale Feature Interaction Module (SSFI) and a CNN-based Cross-scale Feature Fusion Module (CCFM). It is designed to progressively extract and encode feature information through stacked encoder layers, capturing semantic information across various scales and levels while reducing computational cost. The encoded features are then fed into the decoder layers, where they are iteratively extracted by two attention mechanisms: Multi-head attention [51] and Multi-scale deformable attention. These mechanisms enable the extraction of contextually relevant information related to the object position and category. The FFN outputs a set of predicted boxes and corresponding category probabilities. In the following sections, a detailed introduction to the structure of FSH-DETR is provided.

### 3.2. ConvNeXt Backbone

ResNet has been widely used as the backbone for various vision models due to its remarkable performance. Recently, Liu et al. have introduced an improved version of ResNet, called ConvNeXt [52], following an in-depth analysis of the Swin Transformer [53] architecture. The replacement of the original ResNet50 with ConvNeXt-tiny has been demonstrated to achieve enhanced precision and reduced computational cost, while maintaining a comparable number of parameters. The modifications made to ConvNeXt can be divided into two levels: macro and micro.

In terms of macro design, ConvNeXt modifies the stacking ratio of blocks in each stage. The first, second, third, and fourth backbone stages contain, respectively, 3, 3, 9, and 3 blocks. Furthermore, the stem cells in ResNet are replaced with the same patchy layer as Swin Transformer. Additionally, ConvNeXt introduces the concept of group convolution. By dividing the input feature map into multiple subgroups and performing independent convolution operations on each subgroup, the features of different subgroups are fused. This strategy allows the backbone to capture features of different scales. ConvNeXt also adopts the Inverted Bottleneck module to effectively avoid information loss. Finally, a larger convolution kernel is selected to obtain a wider receptive field, thereby improving the ability to perceive global and larger-scale features.

In terms of micro design, ConvNeXt changes the activation functions ReLU and Batch Normalization (BN) to GELU and Layer Normalization (LN), while reducing the number of activation functions and normalization layers. Moreover, ConvNeXt incorporates an LN before and after downsampling to maintain model stability. The aforementioned enhancements ensure that ConvNeXt retains its simplicity while offering faster inference speeds and superior performance compared to the Swin Transformer. Fire and smoke are diverse, with varied flame colors resulting from different fire sources. The size of a fire affects the transparency of smoke, while scene variances, such as interference, concealment, and lighting conditions, can heighten recognition. Most network structures overlook this point. ConvNeXt increases the base channel count from 64 to 96, enabling it to better extract features of fire, smoke, and humans. The aforementioned enhancements confer a natural advantage to ConvNeXt in the domains of fire, smoke, and human detection.

### 3.3. Mixed Encoder

The encoder of Deformable DETR has two functions: implementing deformable attention and feature fusion. These functions have inherently inadequate performance for both tasks. Our solution is the Mixed Encoder, which decouples the original encoder into two modules: Separate Single-scale Feature Interaction (SSFI) and CNN-based Cross-scale Feature Fusion Module (CCFM). The two modules perform self-attention and multi-scale feature fusion respectively.

#### 3.3.1. Separable Single-Scale Feature Interaction

In order to prevent feature fusion from occurring in the encoder, we develop an enhanced module called SSFI. The structures of the original encoder and SSFI are shown in Figure 2a,b, respectively. Although the SSFI architecture appears more complex, resulting in a higher computational cost, it is important to note that this is offset by the reduction in the number of encoder layers. The time complexity of deformable attention and separate self-attention is both $O\left(k\right).$ However, in the original encoder, there are 6 encoder layers, while our Mixed Encoder only contains 2 encoder layers. Furthermore, independently performing self-attention on the outputs of different stages of the backbone also plays an important role in reducing the computational cost, as the self-attention operation is performed on smaller feature maps. Therefore, our Mixed Encoder exhibits a reduced computational cost.

As shown in Figure 2a, the original Deformable DETR flattens and concatenates features from various scales before the encoder to form a long token. Subsequently, it collaborates with Multi-Scale Deformable Attention to standardize the reference points of disparate scale features, thereby enabling their fusion. To avoid this type of feature integration during the encoder stage, we flatten the features at different scales and feed them directly into the encoder without concatenation. This approach results in three different short tokens, which are more readily recoverable. The three short tokens will be independently performed operations, such as separate self-attention and layer normalization, as shown in Figure 2b. Then, the result will be transformed into the state before being flattened at the end of SSFI. Additionally, given that fire, smoke, and human detection models are usually deployed on hardware with limited resources, a streamlined method is specifically adapted for feature interaction at the same scale. In order to replace Multi-scale Deformable Attention in Deformable DETR, we have employed separate self-attention [54]. As an efficient variant of the self-attention mechanism, separate self-attention has the characteristics of low time complexity and latency compared to Multi-head attention in DETR, making it an ideal candidate for deployment on resource-limited hardware. We will provide a more detailed introduction to separable self-attention within SSFI.

The specific pipeline of separate self-attention is shown in Figure 3. When the feature $X\in R^{k\times d}$ is fed into the module, it is directed to three different branches: input I, key K, value V. To convert the k d-dimensional tokens into k scalars, a linear layer is used in the branch I, which essentially multiplies the input X by a weighted matrix $W_{I}\in R^{d\times 1}$ and adds the corresponding bias. The weight $W_{I}$ serves as a latent node L and will be used in subsequent processes. Afterward, scalars are used to form an intermediate variable called context scores through the softmax function. It is worth noting that in the Multi-head attention of the transformer, each input query will calculate a self-attention score with the key, while in the separable self-attention, the key will only calculate the context score with the corresponding latent node L. This crucial operation results in the time complexity of O(k) for separate self-attention, accompanied by a slight decrease in detection precision and a significant decrease in latency [54]. Next, the context score is a broadcasted element-wise multiplication with k d-dimensional vectors that pass through the branch K with a weight of $W_{K}\in R^{d\times d}$, followed by summation to obtain a d-dimensional vector termed the context vector.

Similarly, after passing through branch V, the input X is immediately followed by a ReLU activation function to obtain an intermediate variable $X_{V}\in R^{k\times d}$. The $X_{V}$ then performs broadcasted element-wise multiplication with the context vector and is further processed by a linear layer with a weighted matrix $W_{O}\in R^{d\times d}$ to obtain the final result $y\in R^{k\times d}$. The entire process of separable self-attention can be expressed mathematically as Equation (1):(1)$$y=\left(\underset{c_{v}\in R^{1\times d}}{\underbrace{\sum \left(\overset{c_{s}\in R^{k\times 1}}{\overbrace{\sigma \left(XW_{I}\right)}}*XW_{K}\right)}}*ReLU\left(XW_{V}\right)\right)W_{O}$$
where σ represents the softmax function and ∗ represents the broadcasted element-wise multiplication operation. The calculation of $c_{v}$ can be expressed as Equation (2):(2)$$c_{v}=\sum_{i=1}^{k}c_{s}\left(i\right)X_{K}\left(i\right)$$
where k represents the number of tokens, $c_{s}$ represents context score, and $X_{K}$ represents the output feature of branch K. Equation (2) implements the function of encoding information from all tokens in the input X.

#### 3.3.2. CNN-Based Cross-Scale Feature-Fusion Module

Inspired by Real-Time DETR (RT-DETR) [31], we introduce the CCFM to facilitate feature fusion across different scales. The specific structure of this module is illustrated in Figure 4. The CCFM comprises several fusion modules, each comprising multiple convolutional layers and RepBlocks. These fusion modules facilitate the integration of features across different scales. Low-scale features tend to emphasize global structure and semantic information, whereas high-scale features are more inclined to capture local details and texture information. By enabling the fusion of contextual information, the precision of fire, smoke, and human detection can be enhanced.

The output $r_{2}$, $r_{3}$, and $r_{4}$ of SSFI will serve as the input of CCFM. $r_{4}$ initially passes through the Conv_1×1_ Block shown in Figure 4 and undergoes an upsampling operation. The calculation process of the Conv_1×1_ Block is expressed as Equation (3):(3)$$Conv_{1\times 1}Block=SiLU(BatchNorm(Conv_{1\times 1}(f_{in})))$$
where SiLU is the Sigmoid Gated Linear Unit activation function, BatchNorm represents the batch normalization operation, $Conv_{1\times 1}$ represents the convolution layer with 1×1 kernels, and $f_{in}$ represents the input features.

Subsequently, the output enters the fusion module with $r_{3}$. The fused result undergoes and is sent to upsampling and performs feature fusion with $r_{2}$. Similarly, we replace the Conv_1×1_ Block and upsampling operation with the Conv_3×3_ Block and downsampling operation, repeating the above operation from bottom to top. In the end, the results of feature fusion are concatenated to obtain the final feature. Equations (4) and (5) represent the calculation process of the fusion module and Conv_3×3_ Block, respectively.
(4)$$Conv_{3\times 3}Block=SiLU(BatchNorm(Conv_{3\times 3}(f_{in})))$$
(5)$$Fusion=Flatten(Conv_{1\times 1}\left(cat\left(f_{in1},f_{in2}\right)\right)+RepBlocks\left(Conv_{1\times 1}\left(cat\left(f_{in1},f_{in2}\right)\right)\right))$$
where $Conv_{3\times 3}$ represents using 3 × 3 convolutions to extract features, Flatten represents the flattening operation, cat represents the concatenation operation, RepBlocks indicates RepBlocks, and $f_{in1}$ and $f_{in2}$ represent different input features.

### 3.4. IoU-Based Loss Function

IoU-based loss functions are commonly employed in object detection, quantifying the degree of overlap between predicted and ground truth boxes. Fire and smoke exhibit intricate texture and color attributes, as well as distinctive shapes with unpredictable transformations. Fierce flames and strong smoke can readily obstruct the human body, presenting a significant challenge in detection. Moreover, flaming and smoking from different combustible materials display varying hues and shapes within the same scene, making it difficult for the model to learn complex features and slowing down model convergence. Consequently, the selection of an appropriate IoU-based loss function is of paramount importance. A superior IoU-based loss function facilitates the alignment of the predicted box with the ground truth box in a timely manner, thereby accelerating model convergence. Typically, IoU-based loss functions can be defined as follows:(6)$$L=1-IoU+R\left(A,B\right)$$
(7)$$IoU=\frac{\left|A\cap B\right|}{\left|A\cup B\right|}$$
where A and B represent the predicted box and ground-truth box, respectively. R(·) represents the penalty function. A∩B means the area of intersection between the predicted and ground truth boxes, while A∪B means the area of union between the two bounding boxes.

#### Powerful IoU

Recently, studies by Liu, C. et al. indicated that anchor boxes are prone to expand during the regression process, which seriously affects the convergence speed of the model. Therefore, they proposed PIoU [55]. The formula for $R_{PIoU}$ is as follows:(8)$$P=(\frac{dw_{1}}{w_{gt}}+\frac{dw_{2}}{w_{gt}}+\frac{dh_{1}}{h_{gt}}+\frac{dh_{2}}{h_{gt}})/4,$$
(9)$$f(x)=1-e^{-x^{2}},$$
(10)$$R_{PIoU}=f(P)$$
where $w^{gt}$ and $h^{gt}$ represent the width and length of ground truth box, respectively. The distance between the predicted box and the ground truth box is measured by $dw_{1}$, $dw_{2}$, $dh_{1}$, and $dh_{2}$, and their specific meanings are shown in Figure 5. During the training process, the penalty term P remains constant even if the anchor box expands. This prevents excessive anchor box expansions during regression. Furthermore, the penalty function selected generates an appropriate gradient based on the quality of predicted boxes. When the penalty factor P is greater than 2, signifying a substantial difference between the predicted box and ground-truth box, f′(P) diminishes, thereby mitigating detrimental gradients from low-quality anchor boxes. When P is approximately 1, it indicates proximity between the predicted box and ground-truth box. The f′(P) becomes higher and leads to quicker regression. As P approaches 0, it signifies the predicted box nearing the ground-truth box. *f*′(*P*) gradually decreases as the anchor box’s quality improves, enabling stable optimization towards complete alignment.

PIoU v2 is an extension of PIoU v1, incorporating a non-monotonic attention layer that is controlled by a single hyperparameter. The mathematical formulae are as follows:(11)$$q=e^{-P},\;q\in \left(0,\right.\left.1\right],$$
(12)$$u\left(x\right)=3xe^{-x^{2}},$$
(13)$$L_{PIoU\_v1}=1-IoU+R_{PIoU},$$
(14)$$L_{PIoU\_v2}=u\left(\lambda q\right)L_{PIoU\_v1}$$
where $u\left(\lambda q\right)$ is an attention function. λ is a hyperparameter, and P is the penalty term in PIoU v1. The original penalty term P is replaced by q in PIoU v2. As P increases from 0, *q* gradually decreases from 1, and $u\left(\lambda q\right)$ will undergo a process of initially increasing and then decreasing. $u\left(\lambda q\right)$ reaches its maximum when encountering a medium-quality anchor box. This newly introduced attention mechanism helps the model focus more on medium-quality anchor boxes, reducing the negative impact of low-quality anchor boxes on gradients. A comparison of multiple IoU loss functions reveals that PIoU v2 is the optimal choice, as the traditional IoU loss function treats all anchor boxes equally regardless of their quality. This can lead to suboptimal learning of bounding boxes with varying qualities. PIoU V2 represents a novel approach that combines the strengths of EioU [56], SioU [57], and WioU [58]. It generates a small but increasing gradient for low-quality anchor boxes, allowing them to gradually improve during the regression process. For medium-quality anchor boxes, a large gradient is generated, enabling them to rapidly become high-quality anchor boxes. Medium-quality bounding boxes frequently exhibit overlap with the target but are not perfectly aligned. By directing greater attention to these bounding boxes, PIoU v2 facilitates the model better, learning the position shift and shape transformation of the target. This improves the precision of object localization, resulting in detection boxes that are more closely aligned with the true position of the object. Moreover, PIoU v2 not only reduces the number of hyperparameters but also solves the problem of box expanding during the regression process, which helps to enhance the performance and robustness of the model.

## 4. Experiment Settings

### 4.1. Image Dataset

Our dataset is collected through the internet, including images captured from various sources. It encompasses images captured from a variety of shooting angles, as well as images of different fire morphologies, smoke patterns, and environmental settings. Some of the images in the dataset are shown in Figure 6. During training, all images are resized to 640 × 640 and then subjected to a series of data augmentations, including horizontal and vertical flips, 90-degree rotations, and Salt and Pepper noise. As a result, over 25,000 images are obtained for this experiment. The details of the dataset are provided in Table 1.

### 4.2. Evaluation Metrics

To evaluate the detection performance of different models, we employ Accuracy and AP (Average Precision) as the evaluation metrics. Accuracy is calculated by counting the true positives, and AP is the enclosed area of the PR curve. Specifically, the precision and recall can be computed by the following:(15)$$Precision=\frac{TP}{TP+FP}$$
(16)$$Recall=\frac{TP}{TP+FN}$$
where TP, FP, and FN represent the True Positive, False Positive, and False Negative, respectively. We also use ${AP}_{s}$, ${AP}_{M}$, and ${AP}_{l}$ to represent the AP of small, medium, and large objects, respectively, whereas ${AP}_{50}$ stands for the AP in the case of IoU = 0.5. mAP represent the mean AP of all classes.

In terms of model complexity, we use Giga Floating-point Operations (GFLOPs) to evaluate the computational cost of the model. In addition, the parameter quantity (Params) is used to measure the computational complexity. The larger GFLOPs and Params, the higher the hardware requirements.

To evaluate the inference (prediction) speed of the model, the Frame Per Second (FPS) is employed. A larger FPS means that the model can process more frames per second, which indicates better efficiency of the model.

### 4.3. Experimental Environment

All experiments are conducted on a computer equipped with 4 RTX 3090 GPUs, the CUDA version is 11.7, and the Python version is 3.8. We implement our model using the Pytorch 1.11 [59] and MMDetection [60] framework.

### 4.4. Optimization Method and Other Details

The specific parameter configurations are presented in Table 2. The batch size per GPU is 4 and the total batch size is 16. Besides, the AdamW optimizer [61] is adopted with a base learning rate of 0.0002 and weight decay of 0.0001.

## 5. Result Analysis

### 5.1. Effectiveness of Backbone

To demonstrate the effectiveness of the backbone, we take several mainstream backbone architectures to compare with our ConvNeXt-tiny, including ResNet, EfficientNet [62], and ConvNextv2 [63], to extract features from the input images. We train and evaluate our model with different backbone architectures while keeping other hyperparameters and training procedures consistent. The detection results of the baseline under different backbones are shown in Table 3. The results demonstrate that the choice of backbone can impact the detection precision of the model. Furthermore, implementing ConvNeXt-tiny as the backbone not only reduces the parameters and computation cost but also significantly enhances the detection precision. Although ConvNeXtv2-A performs well on the COCO dataset, it results in differences on our dataset. This discrepancy may be attributed to differences in the data.

### 5.2. Effectiveness of PIoU v2

In this subsection, we conduct experiments to verify the effectiveness of PIoU v2 by comparing it with other IoU-based loss functions, including GIoU, DIoU (Distance IoU) [64], CIoU (Complete IoU) [64], and SIoU. The experimental results are presented in Table 4. It can be observed from these results that PIoU v2 can improve the detection precision.

### 5.3. Comparison with Other Models

To demonstrate the overall performance of our method, we compare it with existing representative object-detection algorithms, including YOLO v3 [15], YOLO v5, YOLO v7 [18], YOLO v8, RTMDet [65], DETR, Deformable DETR, Conditional DETR [48], DAB-DETR [49], and Group-DETR [66]. By benchmarking our results against these approaches, we gain insights into the advancements achieved by our proposed method. All the experiments are performed on the dataset that is introduced in Section 4.1. The results are presented in Table 5, with the best results highlighted in bold. According to the results, FSH-DETR achieved the highest mAP among all algorithms. Moreover, other indicators of our method also exceed other DETR-series algorithms. In small-scale object detection, our method delivers impressive results that are only second to RTMDet. Furthermore, in large-scale object detection, its ${mAP}_{l}$ reaches 71.6%, outperforming all other algorithms.

### 5.4. Ablation Experiments

We aim to comprehensively analyze and evaluate the overall performance of our proposed model through a series of ablation experiments. Table 6 presents the results of multiple ablation experiments, where √ denotes that relevant improvement methods have been applied to the baseline, while × denotes that no relevant improvement methods have been applied.
(1)The results of the first and second groups of experiments indicate that ConvNeXt significantly reduces the number of parameters in comparison to the other models while improving ${Accuracy}_{fire}$, ${Accuracy}_{smoke},\;{Accuracy}_{human}$, and mAP.(2)The results of the first and third groups of experiments indicate that upgrading the original encoder to the Mixed Encoder reduces the computational cost but increases the number of parameters and reduces ${Accuracy}_{smoke}$ and ${Accuracy}_{human}$ slightly.(3)The results of the sixth and seventh groups of experiments indicate that although the Mixed Encoder is the main reason for the increase in the model parameter count, it also ensures the improvement in the model’s precision in detecting fires and humans, as well as mAP.(4)The results of the first and fourth experimental groups indicate that using PIoU v2 as the loss function slightly improves the detection precision of the algorithm but has almost no effect on the parameter and computational cost.

### 5.5. Visualization

To better understand the effectiveness of PIoU v2, we visualize the training process using different IoU loss functions. It is worth noting that the pre-trained model provided by MMDetection is used for parameter initialization. Therefore, the mAP of the model does not increase from 0 in the early stages of training. From Figure 7, it is evident that the model using PIoU v2 as the loss function has a faster convergence speed, while DIoU has the slowest convergence speed. After 50 epochs, all IoUs tend to converge and have roughly the same precision. However, PIoU v2 achieves a slightly higher mAP than other models.

To provide a more intuitive demonstration of the superiority of our algorithm, we selected detection results from various scenarios and presented them in Figure 8. We use green, yellow, and blue for indicating fire, smoke, and human, respectively. In the dark scene, our FSH-DETR algorithm performs better than other algorithms by detecting more targets and with more accurate boxes. In the bright scene, FSH-DETR also detects more small-scale targets than other algorithms.

## 6. Discussion

### 6.1. Limitions

The detection of fire, smoke, and humans is a highly challenging task in object detection. In the actual detection process, the presence of false smoke or fire, such as clouds, steam, and halogen lamps, can pose significant challenges to the detection task. These challenges are further compounded in special environments, such as foggy weather and low-light environments, which further increase the difficulty of detection. Despite the introduction of the Mixed Encoder, which is a module with enhanced fusion capabilities for fire, smoke, and human features, the aforementioned issues remain unresolved. Furthermore, although our proposed FSH-DETR has a higher frame per second (FPS) compared to the baseline, it has not yet met the requirements for real-time monitoring on edge computing devices.

### 6.2. Potential Future Work

In future work, we intend to enhance the dataset through the use of generative adversarial networks (GANs) and diffusion models, which can generate negative samples. This will improve the model’s ability to detect fake fire and smoke. In addition, we posit that the attention mechanism can be employed to further extract features of fire and smoke, thereby assisting detectors in more effectively distinguishing between genuine and spurious instances of fire and smoke. In light of the fact that DETR is still a novel technology, several avenues for future work can be developed based on the findings of this study. One avenue for future work is to extend our approach to real-time video-based fire and smoke detection. The objective is to enhance the real-time processing capability of the model while reducing its computational complexity, thereby ensuring its effectiveness. This will facilitate the development of practical applications.

## 7. Conclusions

The rapid development of deep learning technology has led to an increased use of object-detection techniques in fields, such as forest fire surveillance, fire emergency identification, and industrial safety. Nevertheless, there is still considerable scope for further improvements in this technology. The proposed model, FSH-DETR, employs the advanced Deformable DETR as a baseline to accurately identify and localize instances of fire, smoke, and humans in images. The employment of ConvNeXt, due to its powerful ability and lightweight design, enables the FSH-DETR to extract richer and more comprehensive feature information. Subsequently, the Mixed Encoder, comprising SSFI and CCFM modules, is developed. This approach reduces the computational cost while maintaining high precision. Finally, we introduce the latest PIoU v2, which not only accelerates the convergence speed and improves its robustness in complex fire scenarios, but also raises the detection precision to a new level. Extensive experimentation and evaluation have demonstrated the effectiveness and potential of our approach. The model ultimately achieves a mAP of 66.7%, outperforming the comparative models.

## Figures and Tables

**Figure 1 sensors-24-04077-f001:**
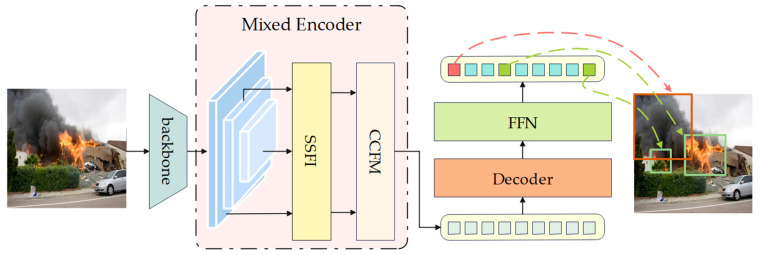
The overall architecture of FSH-DETR.

**Figure 2 sensors-24-04077-f002:**
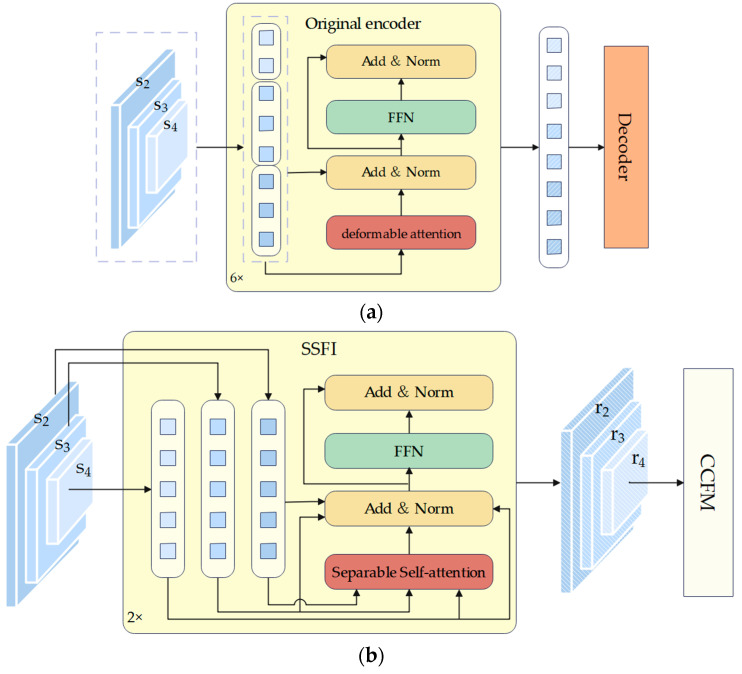
Architecture of original encoder and SSFI. (**a**) Architecture of original encoder in baseline. (**b**) Architecture of SSFI in FSH-DETR.

**Figure 3 sensors-24-04077-f003:**
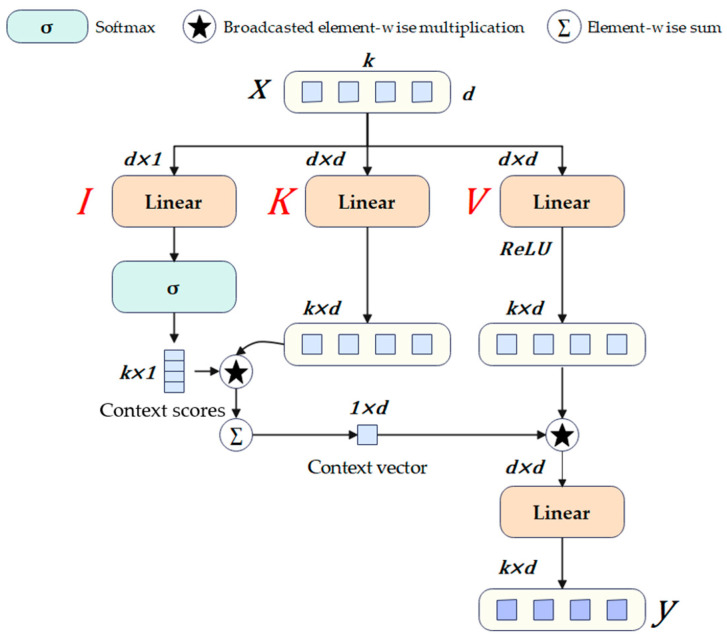
The architecture of the separable self-attention block.

**Figure 4 sensors-24-04077-f004:**
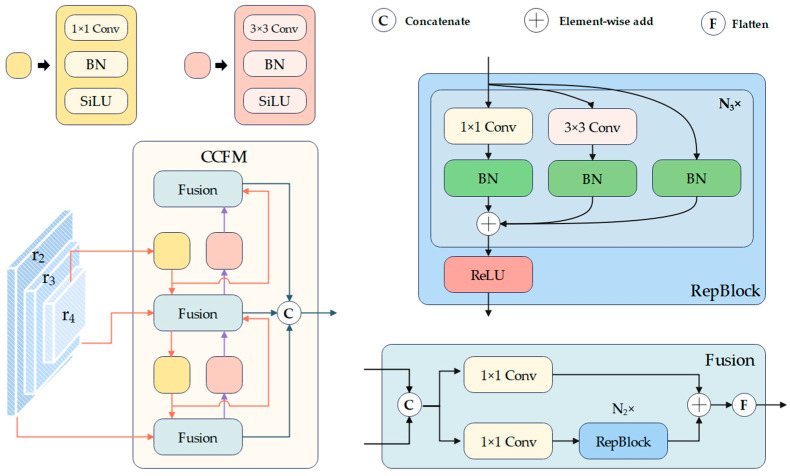
Architecture of CCFM.

**Figure 5 sensors-24-04077-f005:**
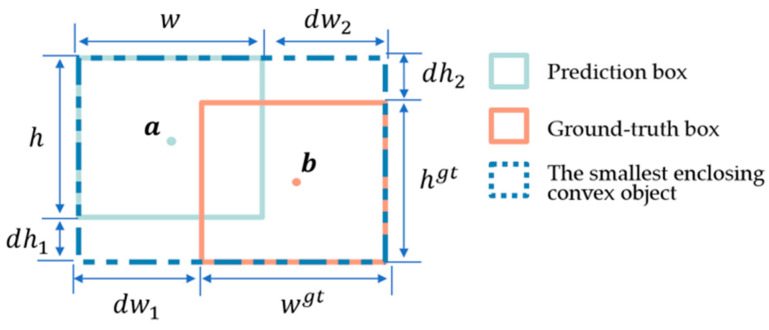
Schematic of loss function parameters.

**Figure 6 sensors-24-04077-f006:**
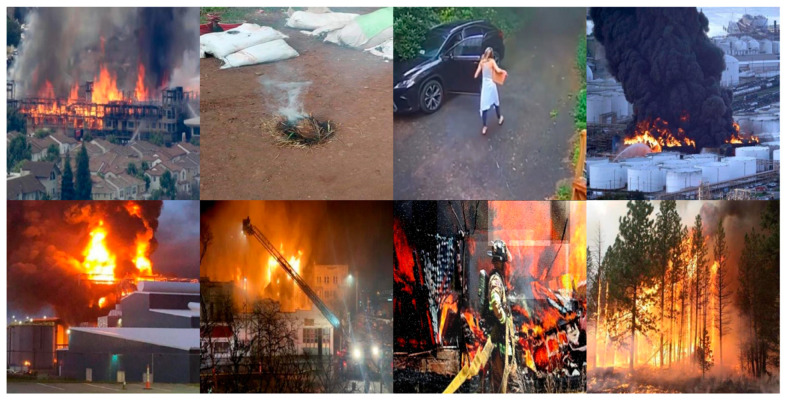
Sample images of the collected dataset.

**Figure 7 sensors-24-04077-f007:**
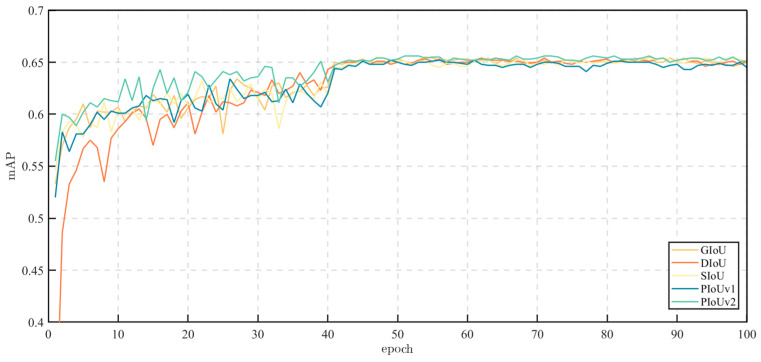
The training process curve of the baseline under different IoU-based loss functions.

**Figure 8 sensors-24-04077-f008:**
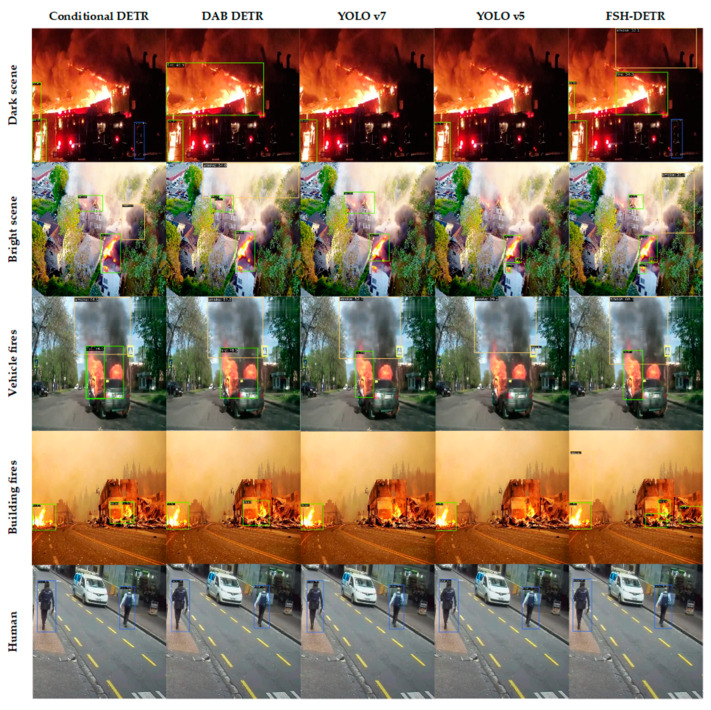
The detection performance of FSH-DETR and other algorithms under different situations.

**Table 1 sensors-24-04077-t001:** Fire smoke and human dataset and its specification.

Dataset	Number of Images	Fire Objects	Smoke Objects	Human Objects
Train	20,016	21,809	14,896	11,568
Evaluation	5004	8135	4000	2175
Total	25,020	29,944	18,896	13,743

**Table 2 sensors-24-04077-t002:** Parameter configurations in the experiment.

Parameter Name	Parameter Value
epoch	100
batch size	16
optimizer	AdamW
learning rate	0.0002
weight decay	0.0001

**Table 3 sensors-24-04077-t003:** The performance of Deformable DETR under different backbones. The best results are highlighted in bold.

Backbone	mAP	${mAP}_{50}$	${mAP}_{75}$	${mAP}_{s}$	${mAP}_{m}$	${mAP}_{l}$	GFLOPs	Params (M)	FPS
ResNet-50	65.5	84.0	63.7	45.1	53.7	70.6	126.0	41.1	25.1
EfficientNet-b0	64.9	81.6	63.3	35.6	53.4	69.8	71.3	**16.4**	18.9
ConvNeXtv2-A	60.2	74.4	60.1	27.2	49.4	65.2	74.4	41.9	19.6
ConvNeXt-tiny	**66.1**	**84.3**	**65.2**	**53.6**	**53.1**	**71.6**	**70.8**	40.8	** 29.8 **

**Table 4 sensors-24-04077-t004:** The performance of the baseline under different IoU-based loss functions. The best results are highlighted in bold.

IoU Loss Function	mAP	${mAP}_{50}$	${mAP}_{75}$	${mAP}_{s}$	${mAP}_{m}$	${mAP}_{l}$	Total Training Time (h)
GIoU	65.5	** 84.0 **	63.7	45.1	53.7	70.6	23.2
DIoU	65.4	82.8	63.8	39.1	52.4	70.6	19.2
CIoU	65.6	83.8	64.4	43.2	**54.3**	70.8	**18.0**
SIoU	65.5	83.6	64.6	41.1	53.1	70.6	19.2
PIoUv1	65.2	83.3	64.5	**48.7**	51.7	70.5	18.9
PIoUv2	**65.6**	83.6	**64.8**	48.2	52.8	**70.7**	19.5

**Table 5 sensors-24-04077-t005:** Comparison results between FSH-DETR and other models. The best results are highlighted in bold.

Model	Backbone	mAP	${mAP}_{50}$	${mAP}_{75}$	${mAP}_{s}$	${mAP}_{m}$	${mAP}_{l}$	FPS
YOLOv3	DarkNet-53	57.2	78.3	59.4	36.7	47.0	62.3	68.5
YOLOv5	YOLOv5-n	63.9	79.5	63.1	24.5	52.7	68.8	92.5
YOLOv7	YOLOv7-tiny	65.2	81.3	63.2	33.8	**54.8**	69.6	**93.9**
YOLOv8	YOLOv8-n	64.9	79.0	63.2	33.5	55.3	69.0	64.6
RTMDet	RTMDet-tiny	65.2	79.8	64.1	**59.8**	55.1	69.3	42.2
DETR	R-50	62.6	81.9	62.6	17.3	46.8	68.7	34.1
Deformable DETR	R-50	65.5	84.0	63.7	45.1	53.7	70.6	25.1
Conditional DETR	R-50	64.2	82.6	63.7	27.7	50.2	70.2	30.8
DAB-DETR	R-50	65.1	83.1	65.2	25.1	52.4	70.6	24.9
Group-DETR	R-50	65.6	83.2	64.3	43.5	51.9	71.1	19.3
**Ours**	ConvNeXt	**66.** **7**	**84.** **2**	**65.3**	50.2	54.0	**71.6**	28.4

**Table 6 sensors-24-04077-t006:** Results of ablation experiments on FSH-DETR. √ denotes that relevant improvement methods have been applied to the baseline, while × denotes that no relevant improvement methods have been applied. The best results are highlighted in bold.

Improved Methods			Evaluation Metrics					
ConvNeXt	Mixed Encoder	Loss Function	mAP	${Accuracy}_{fire}$	${Accuracy}_{smoke}$	${Accuracy}_{human}$	GFLOPs	Params (M)
×	×	×	65.5	96.89	73.97	79.88	126.0	41.1
√	×	×	66.1	97.50	80.48	80.17	**70.8**	40.8
×	√	×	65.8	98.01	73.27	79.99	75.5	46.3
×	×	√	65.6	97.21	76.91	78.62	123.0	** 40.1 **
√	√	×	66.6	98.05	78.09	78.89	77.5	50.1
√	×	√	66.2	97.62	**80.75**	79.40	79.8	40.8
√	√	√	**66.** **7**	**98.05**	78.78	**80.22**	77.5	50.8

## Data Availability

Restrictions apply to the datasets.

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
