# Peer review of "FSH-DETR: An Efficient End-to-End Fire Smoke and Human Detection Based on a Deformable DEtection TRansformer (DETR)"

_sensors, 2024, doi:10.3390/s24134077_

Round 1
Reviewer 1 Report
Comments and Suggestions for Authors
1. The abstract is not correlated with the title of the paper. Usually, not recommended to use abbreviations in the title like FCM-DETR, or DETR. The abstract of the manuscript should be revised as the new contributions and the explanation are not clear.
2. Please check the manuscript carefully, for example, what do some abbreviations state for the first time; such as GIoU, CCFM, SSFI, YOLO in the Abstract and etc.
3. Please make a Related Work section with recently published fire detection and recognition papers such as Machine/Deep Learning and other famous approaches. I reviewed from the internet several following smoke detection papers that are expected to help more ideas to readers with previously achieved works.
https://doi.org/10.3390/s23208374
https://doi.org/10.3390/atmos11111241
https://doi.org/10.3390/fire7030084
https://doi.org/10.3390/s22239384
4. Dataset. Please provide a table with the number of datasets used for smoke/fire images as well as other pertinent data.
5. Can the authors provide some insight into the minimum requirements (such the quantity of frames needed) for the training data set? Do you intend to artificially enhance the training dataset using the augmentation technique?
6. Any way to reduce the computational cost compared to other approaches, please discuss.
7. Overfitting resulted in achieved results? In the experiment section, please have a discussion.
8. The suggested method's shortcomings should have been mentioned by the authors, but they didn't. Suggested technique how can one differentiate between fire-like scenes such as sunrise or sunset?
9. Can we identify fake smoke or fires with this system? To put it another way, there are situations when fire-like events or environments are mistakenly labeled as fires and warnings, which is why fire detection methods can occasionally detect a fire even in non-fire photographs. Artificial fires are easily distinguished by human sight, but because of their similar brightness and reflection, computers occasionally mistakenly identify streetlights, neon signs, and car headlights as genuine fires.
Summary: This manuscript's scientific and technical novelty and contribution are not sufficient enough. It seems an incremental work than novel research. The organization is not good, it should be thoroughly revised. The limitations and future work are also missing, please include them. In many places of the work, the style of language is inappropriate for a scientific paper. The representation of results needs further optimization. Some results are not displayed correctly, such as in Tables and Figures. Please address all the comments with high responsibility and actively.
Author Response
Dear editor and reviewers,
Thank you for offering us an opportunity to improve the quality of our submitted manuscript sensors-3034699. We appreciated very much the reviewers' constructive and insightful comments. In this revision, we have addressed all of these comments/suggestions. We hope the revised manuscript has now met the publication standard of your journal.
Please see the attachment. In the attachment, our point-to-point responses to the queries raised by the reviewers are listed.
Kind regards,
Mr. Liang

Reviewer 2 Report
Comments and Suggestions for Authors
Novelty:
The paper addresses challenges in fire and smoke detection, such as slow convergence, poor small target detection, and high computational costs. It proposes FCM-DETR, an end-to-end detection algorithm based on Deformable DETR, which incorporates ConvNeXt to replace Resnet, reduces computation and improves feature extraction. A Mixed Encoder structure is introduced for effective multi-scale feature processing. Additionally, a new loss function, Powerful IoU v2, is applied to enhance convergence speed and precision. In my point of view, the paper is not much technically strong as several papers are already on the cloud with similar kinds of solutions. Meanwhile, I have some doubts which need to be addressed.
Methodology:
· In section 3.1, line 237–240240, why suddenly decoder got skipped while explaining?
· Algorithm 1. is making no sense as there is no explanation can be seen in the paper. Also, use proper way of writing it so that the reader can understand it.
· Section 3.2 explains about backbone. As As mentioned in abstract this backbone reduces the complexity of the model, How so? Need explanation about how it is reducing the complexity.
· Section 3.2 line 252, what is 1:3:3:1. It would be better if author explained it in the manuscript.
· Section 3.3 is explaining mixed encoder, this manuscript mixed SSFI and CCFM. What about model’s over all complexity? Don’t you think this is making your model more complex. And if it is reducing the complexity then you should explain how?
· Section 3.3.1 line 296 and 297 “flatten the features at different scales and feed them directly into the encoder without concatenating them. This approach makes it easier to recover tokens later”, but in the manuscript it cannot be seen where you use these feature later?
· Section 3.3.1 line 320-322, author mention that the complexity reduction meanwhile small decrease in accuracy and much in latency, but there is no proper proof and explanation that how it is done?
· Better to explain equation 1, like why you need it and how you got .
· From this equation, it can be clearly seen that the context score is important to compute y, but how it got computed through equation 1 is not clearly explained.
· Section 3.3.2, as it is one the major contribution of the paper, it should be explained clearly and well but this version is lacking it.
· Section 3.3.2 line 349-353, very confusing for the reader. Make it clear what author want to say in it. (How so).
· Overall, it is recommended to use same math notations instead of different.
· All the loss functions explained in the manuscript are not the part of the contributions so no need to explain about them.
· It would be better if Algorithm is provided to explain how the experiment is designed.
· All these evaluation matrices are well known and not your contribution, so no need to give this much explanation.
Results:
· Table 3 “The performance of Deformable DETR under different backbones”, why D-DETR, why not your own model. As author has to proof his model efficiency then perform this experiment on your model instead of baseline.
· Similarly, for Table 4. These loss functions are already proven on these baselines, no need to prove them again here. It is better to show your own model’s performance in this aspect.
Writing:
· Overall, very poor paper presentation, need to be improved so that reader can make intrest to read it and gets possible benefits.
· The paper contains numerous grammatical errors and generally poor English. A thorough grammar check and revision are necessary.

The paper contains numerous grammatical errors and generally poor English. A thorough grammar check and revision are necessary.
Author Response
Dear editor and reviewers,
We would like to begin with our sincere appreciation for all the valuable comments, insightful suggestions and thoughtful corrections offered by the reviewers and editor to our manuscript sensors-3034699. The comments and suggestions definitely helped us to improve the quality of the manuscript. We have revised the manuscript in which major changes are highlighted in red. These changes are summarized in the attachment. Please see the attachment.
Kind regards,
Mr. Liang

Round 2
Reviewer 1 Report
Comments and Suggestions for Authors
Current version of the paper was improved and meets publication requirements.
Author Response
Thank you to the editor and reviewer for your detailed comments and suggestions. Your input has been invaluable in improving the quality of our manuscript.
Reviewer 2 Report
Comments and Suggestions for Authors
-- English is still poor, It can be revised properly.
-- See my previous comment, as some equations still need explanation.
-- Quality of the figures can be approved.
Comments on the Quality of English LanguageIt should be approved.
Author Response
Comment 1: English is still poor; It can be revised properly.
Response: Thank you for your valued suggestion.
We have made significant language polishing to the manuscript, striving to present clear experimental ideas and eliminate confusion for readers when reading. The modifications are highlighted in red.
Comment 2: See my previous comment, as some equations still need explanation.
Response: Thank you for your comment and suggestion.
We have provided a more detailed explanation of some equations. The added content is highlighted in blue.
In Page 12:
means the area of intersection between the predicted and ground truth boxes, while means the area of union between the two bounding boxes.
In Page 13:
When the penalty factor P is greater than 2, signifying a substantial difference between the predicted box and ground-truth box, f’(P) diminishes, thereby mitigating detrimental gradients from low-quality anchor boxes. When P is approximately 1, it indicates proximity between the predicted box and ground-truth box. The f’(P) becomes higher and leads to quicker regression. As P approaches 0, it signifies the predicted box nearing the ground-truth box. ?'(?) gradually decreases as the anchor box's quality improves, enabling stable optimization towards complete alignment.
In Page 13:
As P increases from 0, ? gradually decreases from 1, and u(λq) will undergo a process of initially increasing and then decreasing. u(λq) reaches its maximum when encountering a medium-quality anchor box.
